# Species-Specific Seasonal Shifts in Reproductive Allocation in the Southern Grass Lizard, *Takydromus sexlineatus* (Lacertidae)

**DOI:** 10.3390/ani14081167

**Published:** 2024-04-12

**Authors:** Cai-Feng Wang, Yu Du, Kun Guo, Xiang Ji

**Affiliations:** 1Zhejiang Provincial Key Laboratory for Water Environment and Marine Biological Resources Protection, College of Life and Environmental Sciences, Wenzhou University, Wenzhou 325035, China; wangcaifeng@stu.wzu.edu.cn (C.-F.W.); guokun8808@wzu.edu.cn (K.G.); 2Institute for Eco-Environmental Research of Sanyang Wetland, Wenzhou University, Wenzhou 325014, China; 3Hainan Key Laboratory of Herpetological Research, College of Fisheries and Life Science, Hainan Tropical Ocean University, Sanya 572022, China; yudu@hntou.edu.cn

**Keywords:** clutch size, clutch mass, egg size–number trade-off, Lacertidae, egg size, female reproduction, life history variability

## Abstract

**Simple Summary:**

We designed a common garden experiment to collect data on female reproductive traits from three populations of *Takydromus sexlineatus*, testing the hypothesis that geographically separated populations should share a species-specific pattern of seasonal shifts in reproductive allocation. Six traits differed among populations, with four of the six also differing among successive clutches. Females grew longer during the breeding season and produced more eggs in the first clutch than in the subsequent clutches; egg size was unchanged throughout the breeding season. After removing the influence of female size or postpartum mass we found that (1) postpartum body mass, clutch mass, and relative clutch mass were greater in the Wuzhishan population than in the Shaoguan and Zhaoqing populations; (2) egg size was greatest in the Wuzhishan population and smallest in the Zhaoqing population; and (3) clutch size was greatest in the Wuzhishan population and smallest in the Shaoguan population. Our data validate the hypothesis tested.

**Abstract:**

We designed a common garden design to collect data on female reproductive traits from three populations of the southern grass lizard *Takydromus sexlineatus*, testing the hypothesis that a species-specific pattern of seasonal shifts in reproductive allocation should be shared by geographically separated populations. Of the seven examined traits, six differed among populations, with four of the six also differing among successive clutches. Females grew longer during the breeding season and produced more eggs in the first clutch than in the subsequent clutches; egg size was unchanged throughout the breeding season. After removing the influence of female size or postpartum body mass we found the following. First, postpartum body mass, clutch mass, and relative clutch mass were greater in the Wuzhishan population than in the Shaoguan and Zhaoqing populations. Second, egg size was greatest in the Wuzhishan population and smallest in the Zhaoqing population. Third, clutch size was greatest in the Wuzhishan population and smallest in the Shaoguan population. Females did not trade-off egg size against number within each population × clutch combination. Our study validates the hypothesis tested, supports the conventional view that reproductive output is highly linked to maternal body size in lizards, and follows the classic prediction that females with different amounts of resources to invest in reproduction should give priority to adjusting the total number rather than size of their offspring.

## 1. Introduction

Reproductive traits include the timing and duration of the breeding season, the size range of reproductive individuals, total investment per reproductive episode or season, reproductive frequency (the number of clutches or litters produced per breeding season), and reproductive allocation to offspring (egg or neonate) size versus number. These variables can vary between geographically separated populations, especially those of widespread species adapting to different habitats [1,2,3]. Previous studies generally conclude that patterns of geographical variation in reproductive traits are jointly determined by two mutually non-exclusive mechanisms, phenotypic plasticity and local adaptation, of which the relative importance on individual traits in question can be disentangled by using a reciprocal transplant or common garden design [4,5,6,7]. Many studies have investigated geographical variation in reproductive traits in a diverse array of animal taxa, including lizards of the families Agamidae [8,9], Dactyloidae [10,11], Gekkonidae [12], Lacertidae [13,14,15,16,17,18,19,20], Phrynosomatidae [21,22,23,24,25,26,27,28,29], Polychrotidae [30], Scincidae [31,32,33,34], and Xenosauridae [35]. However, these studies have rarely covered the whole breeding season to compare reproductive traits among populations. For the species where females reproduce multiple times per breeding season, geographical variations in reproductive traits cannot be accurately, objectively, and comprehensively presented without taking seasonal shifts in reproductive allocation into account.

Many lizards of the family Lacertidae are multiple-clutched species [13,14,15,16,17,18,19,20]. Reproductive investment per episode and its allocation to offspring number versus size are not fixed in lacertid lizards but vary seasonally, even when the energy and maternal abdominal space available to hold the clutch have unlimited availability. For example, female common wall lizards (*Podarcis muralis*) produce more and larger eggs in the first clutch and fewer and smaller eggs in the subsequent clutches [36]; female white-striped grass lizards (*Takydromus wolteri*) produce more eggs in the first clutch and fewer eggs in the subsequent clutches but keep egg size unchanged between successive clutches in a breeding season [15]; and female northern grass lizards (*Takydromus septentrionalis*) produce more and smaller eggs in the first clutch and fewer and larger eggs in the subsequent clutches [37]. The Mongolian racerunner *Eremias argus* provides a case where females from a cold-climate population produce more and smaller eggs in the first clutch and fewer and larger eggs in the subsequent clutches [38], whereas their warm-climate conspecifics produce smaller eggs in the first clutch and larger eggs in the subsequent clutches but keep clutch size unchanged between successive clutches throughout the breeding season [39]. What can be inferred from the above examples is that patterns of seasonal shifts in reproductive allocation to offspring size versus number differ among species and even among climatically distinct populations of the same species. However, as studies comparing reproductive traits among populations are far sparser than those of within-population comparisons, a broader collection of data from multiple populations of more species is needed to elucidate not only the life-history consequences of reproductive allocation but also the adaptive significance of seasonal shifts in reproductive investment in individual species.

Here, we have used a common garden design to collect data on seven female reproductive traits [female body size, postpartum body mass, egg size (mass), within-clutch variability in egg mass, clutch size, clutch mass, and relative clutch mass] from three populations of the southern grass lizard *Takydromus sexlineatus*, paying particular attention to the traits varying among populations and between successive clutches in a breeding season. Data from multiple populations are useful in identifying whether the patterns of seasonal shifts in reproductive allocation observed in individual species are species-specific or population-specific patterns. We hypothesize that a species-specific pattern should be shared by geographically separated populations, whereas a population-specific should differ among geographically separated populations.

## 2. Materials and Methods

### 2.1. Study Species

*Takydromus sexlineatus* is a small oviparous lacertid lizard described by Daudin (1802) [40]. The lizard is basically a tropical species that can be found in the southeastern part (Guangxi, Guizhou, Fujian, Hainan, Hong Kong, Hunan, Guangdong, and Yunnan) of Chinese mainland; it also occurs in Burma, Cambodia, Laos, Thailand, Vietnam, the Malaysian Peninsula, and Indonesia (Borneo, Java, Sumatra and Natuna islands) [41]. Despite its wide distribution and the fact that it is phylogenetically and taxonomically well known, the biology and ecology of *T. sexlineatus* remain poorly studied. Earlier studies of *T. sexlineatus* conclude the following: (1) prolonged exposure to temperatures lower than 6 °C or higher than 42 °C is lethal to the lizard, and adults in thermal gradients prefer to maintain their body temperatures within the range from 28 °C to 34 °C, with a mean of 31.5 °C; (2) the body temperature that maximizes locomotor performance (sprint speed) is higher in *T. sexlineatus* (~33 °C) compared with its congeneric counterparts (~28 °C in *T. wolteri* and ~31 °C in *T. septentrionalis*) with more northerly distributions; (3) *T. sexlineatus* is more vulnerable to climate change than *T. wolteri* and *T. septentrionalis* due to the lack of the mechanisms adopted by the latter two species to enhance thermal acclimatization capacity; and (4) the mean incubation length at a given temperature is longest in *T. sexlineatus* and shortest in *T. wolteri*, with *T. septentrionalis* in between [42,43,44].

### 2.2. Animal Collection and Care

We collected adult males and females with various-sized yolking follicles in late March 2015 and 2016 from three localities (populations): one in Wuzhishan (18°50′ N, 109°31′ E), Hainan Island (Province), and the other two in Shaoguan (24°41′ N, 113°48′ E) and Zhaoqing (23°03′ N, 112°27′ E), Guangdong Province (Figure 1). The annual mean temperature is highest in Wuzhishan (22.4 °C) and lowest in Shaoguan (20.9 °C), with Zhaoqing (22.1 °C) in between. The lizards collected in each year were transported to our laboratory where they were sexed and marked via unique combinations of clipped toes for later identification. Following a previous study on *P. muralis* on mating purpose [36], we housed 9–12 lizards (with a female-to-male ratio of approximately 2 to 1) from the same population in each 900 × 650 × 350 mm (length × width × height) communal cage with a moist soil substrate (~50 mm depth), grasses, and pieces of clay tile that served as shelters. We placed the communal cages in a room where temperatures varied from 22 °C to 28 °C. Thermoregulatory opportunities were provided daily between 07:00–19:00 h by a 60 W full-spectrum lamp suspended at one end of each communal cage. Mealworms (*Tenebrio molitor*), house crickets (*Achetus domestica*), and water enriched with multivitamins and minerals were provided ad libitum. We palpated females at 3 d intervals for the presence of eggs in the oviduct. Females with shelled oviductal eggs were removed from the communal cages and housed individually in 200 × 200 × 200 mm egg-laying cages with moist soil (~40 mm depth). In no case did any female remain in an egg-laying cage for >48 h.

We collected, weighed, and measured eggs within 6 h post-laying. After recording the egg-laying date, snout-vent length (SVL), and body mass, we moved postpartum females back to the communal cages, where they remained until they again carried shelled oviductal eggs and, at that time, they were once again transferred to the egg-laying cages. Females were allowed to be fertilized and produce as many clutches as they could in the laboratory. We incubated the eggs under multiple thermal conditions, and we will report data on the hatching success, incubation length, and temperature-induced variation in the hatchling phenotypes elsewhere. We released the lizards collected in each year at their point of capture in late August soon after the egg-laying season.

### 2.3. Data Analyses

Clutch size, clutch mass, clutch mean egg mass (hereafter, egg mass), relative clutch mass (RCM), and within-clutch variability in each egg mass were calculated for each clutch. Clutch size was the total number of eggs in a clutch, clutch mass was the total mass of eggs in a clutch, RCM was calculated by dividing clutch mass by postpartum female mass, and within-clutch variability in egg mass was calculated as the coefficient of variation [CV = 100 × (standard deviation/clutch mean egg mass)].

All statistical analyses were performed with Statistica 10.0 (Tulsa, OK, USA). Before parametric analyses, data were tested for normality using the Kolmogorov–Smirnov test and for homogeneity of variances using Bartlett’s test. We used linear regression analysis to examine the relationship between a selected pair of dependent and independent variables and ANCOVA to examine the slope homogeneity of regression lines. We used data on the first three clutches with large enough sample sizes (≥18) to examine geographic and seasonal variation in female reproductive traits. More specifically, we used a two-way ANOVA (female SVL, egg mass, and CV of egg mass, of which the latter two were found to be independent of female SVL) or ANCOVA (postpartum body mass, clutch size, and clutch mass using female SVL as the covariate, and relative clutch mass using postpartum body mass as the covariate), with the population origin and clutch order as the fixed factors, to examine whether these female reproductive traits differed among populations and among successive clutches. Tukey’s *post hoc* test was performed on a trait that differed among populations and/or clutches. We used partial correlation analysis to examine whether females traded egg size (mass) off against egg number while holding female size (SVL) constant. Throughout this paper, values are presented as mean ± standard error (SE) and range, and the significance level is set at *p* = 0.05.

## 3. Results

Body sizes ranged from 49.6 to 64.2 mm SVL in Wuzhishan females, 44.5 to 57.9 mm SVL in Shaoguan females, and 46.3 to 61.4 mm SVL in Zhaoqing females (Table 1). Females from the three populations laid eggs in the months from April to August. Wuzhishan and Shaoguan females laid up to five clutches and Zhaoqing females laid up to four clutches per egg-laying season, with no females in either population laying more than four eggs per clutch (Table 1).

Data on the first three clutches show that the mean values for female SVL differed among the three populations and among the first three clutches, and that the interaction between population (origin) and clutch (order) was not a significant source of variation in female SVL (Table 2). More specifically, the mean SVL was greater in Wuzhishan females than in females from the other two populations, and it was greater in females laying the subsequent clutches than in females laying their first clutch.

From a series of linear regression analyses within each population × clutch combination, we knew the following. First, postpartum body mass, clutch size, and clutch mass were positively related to female SVL (all *p* < 0.05). Second, egg mass and within-clutch variability in egg mass (CV of egg mass) were independent of female SVL (all *p* > 0.05). Third, clutch mass was positively related to postpartum female mass (all *p* < 0.05). Postpartum body mass differed among the three populations but not among the first three clutches after accounting for female SVL, with the SVL-adjusted mean mass being greater in Wuzhishan females than in females from other two populations (Table 2). Clutch size and clutch mass differed among the three populations and among the first three clutches after accounting for female SVL, and so did relative clutch mass after accounting for postpartum mass (Table 2). The SVL-adjusted mean clutch size was greatest in Wuzhishan females and smallest in Shaoguan females, with Zhaoqing females in between; the SVL-adjusted mean clutch size was greater in the first clutch than in the subsequent two clutches (Table 2). The mean values for clutch mass and RCM were greater in Wuzhishan females than in females from the other two populations and were greater in the first clutch than in the subsequent two clutches after accounting for female SVL or postpartum body mass (Table 2). The mean egg mass was greatest in Wuzhishan females and smallest in Zhaoqing females, with Shaoguan females in between; the mean egg mass did not vary among the first three clutches (Table 2). The within-clutch variability in egg mass remained remarkably consistent across the three populations and the first three clutches (Table 2). Females did not trade-off egg size against egg number, as was revealed by the fact that the negative correlation between egg mass and clutch size was not significant within each population × clutch combination after controlling for female SVL (partial correlation analysis; all *r* > −0.34 and all *p* > 118). Female postpartum mass was a significant determinant of egg mass (linear regression analysis, all *p* < 0.05) but only explained a very small proportions (from 3.8% in the Zhaoqing population to 7.0% in the Wuzhishan population) of variation in the egg mass within each population (Figure 2). After accounting for female postpartum mass, we once again found that the mean egg mass was greatest in Wuzhishan females and smallest in Zhaoqing females, with Shaoguan females in between (Tukey’s post hoc test, all *p* < 0.01).

## 4. Discussion

Of the seven female reproductive traits examined in this study, six differed among the three populations, and four of the six differed among the first three clutches (Table 2). The CV of egg mass was the only trait that did not differ among three populations, nor among the first three clutches. This finding suggests that female southern grass lizards from different populations and in different seasons tend to produce eggs of similar size in the same clutch. Such a strategy of reproductive investment per offspring is often adopted by species existing in fairly constant or predictable environments, following the prediction from the parent–offspring conflict theory for maternal investment in individual offspring [45,46]. The interaction between population origin and clutch order was not a significant source of variation in any examined female reproductive trait (Table 2). This finding is of particular interest because it suggests that the three populations of *T. sexlineatus* share the same patterns of seasonal shifts in all the examined female reproductive traits, thus validating the hypothesis tested. Our finding that reproductive female *T. sexlineatus* grew longer (SVL) but kept postpartum body mass unchanged between successive clutches provides evidence for the existence of two mechanisms of energy allocation during the breeding season. First, as in other animal taxa [47,48,49,50], somatic growth has a higher priority over reproduction in energy allocation in *T. sexlineatus*. Second, like other multiple-clutched lacertid females [15,36,37,38,39], female *T. sexlineatus* maximize annual fecundity by channeling surplus energy not used in the current reproductive episode into the next clutch. It is worth noting that the increases in SVL and optimal reproductive allocation along this study were achieved in females having ad libitum conditions of food and thermal requirements, which are very unlikely to be fulfilled in the field due to various limitations in resource acquisition.

Like female *T. wolteri* [15], female *T. sexlineatus* produced more eggs in the first clutch and fewer eggs in the subsequent clutches but kept egg mass unchanged between successive clutches (Table 2). This suggests that, although these two congeneric species are distributed in climatically different regions (*T. wolteri* in the temperate region, and *T. sexlineatus* in the tropics), they share the same patterns of seasonal shifts in clutch size and egg size, which differ from the patterns recorded in *T. septentrionalis* where females produce more and smaller eggs in the first clutch than in the subsequent clutches [37]. Of these three *Takydromus* species, only *T. septentrionalis* is endemic to China, with its distributional range overlapping south with *T. sexlineatus* and north with *T. wolteri* [41]. Therefore, the above comparisons provide an inference that seasonal shifts in clutch size and egg size do not necessarily vary among congeneric species in a geographically continuous way.

Body condition is crucial to the onset of vitellogenesis in squamate reptiles, with poorly conditioned females often reducing the number of offspring produced per episode or even skipping current reproduction [37,51,52]. Therefore, one plausible explanation for the larger SVL-adjusted mean clutch size in Wuzhishan females lies in their better body conditions, as was revealed by the fact that postpartum body mass was greater in Wuzhishan females than in females from the other two populations after accounting for SVL (Table 2). Alternatively, it might be possible that body conditions during the breeding season were always above the lower threshold required to initiate vitellogenesis of a clutch predicted by using female SVL in each population of *T. sexlineatus*.

We found that the SVL-adjusted mean clutch mass varied among the three populations of *T. sexlineatus* (Table 2). This variation resulted primarily from geographical variation in reproductive investment per episode and its allocation to egg number versus size, as was revealed by the fact that clutch mass, clutch size, and egg size all differed among the three populations (Table 2). RCM is determined by using postpartum female mass and clutch mass, both of which may change geographically and temporally in squamate reptiles [15,33,53]. With all else being equal, RCM should be greater in species where females produce heavier clutches but leave a smaller amount of energy reserves after laying, and geographical variation in RCM should be more apparent in widespread species living in contrasting environments [15].

It is common among reptiles that the total energy allocated to reproduction per episode is positively related to female size; however, reproductive investment per offspring varies both within and among species [52]. Larger female *T. sexlineatus* laid larger and heavier clutches, primarily by producing more eggs rather than larger eggs, as was revealed by the following two facts. First, egg mass was independent of female SVL within each population × clutch combination. Second, egg mass did not vary among successive clutches. Our study is the first to demonstrate that female postpartum body mass explains a significant proportion of variation in egg size in lizards (Figure 1). SVL-adjusted mean postpartum body mass was greater in Wuzhishan females (Table 2), suggesting that they accumulated a greater amount of energy throughout the breeding season and were therefore more likely to produce larger eggs. However, factors other than postpartum body mass also affect egg size in *T. sexlineatus*, as was revealed by the fact that the three populations differed from each other in mean egg mass after accounting for female postpartum mass (Figure 1). Presumably, the finding that the three populations of *T. sexlineatus* differ from each other in mean egg mass can be explained with respect to local adaptation of life history because organisms cannot better adapt to local environments without adjusting their life history traits plastically and/or genetically.

## 5. Conclusions

Larger females of *T. sexlineatus* laid more eggs and heavier clutches. This finding supports the conventional view that fecundity and reproductive output (clutch or litter mass) are highly linked to female body size in lizards. Egg size was less variable than clutch size in *T. sexlineatus*. Therefore, our study follows Smith and Fretwell’s (1974) [49] prediction that females with different amounts of resources to invest in a given reproductive episode are more likely to adjust the number rather than the size of their offspring. Female *T. sexlineatus* did not trade-off egg size against number, suggesting that the lizard is among species where egg size is independent of clutch size. Egg size was independent of female body size but was positively related to female postpartum body mass within each of the three populations of our study. These findings suggest that maternal body condition rather than body size is a determinant of egg size in *T. sexlineatus*. The three populations of *T. sexlineatus* differed from each other in mean egg mass after accounting for female postpartum body mass. This finding is of interest because it suggests that factors other than maternal body condition may also affect egg size in *T. sexlineatus*. Our data validate the hypothesis tested, which states that a species-specific pattern of seasonal shifts in reproductive allocation should be shared by all geographically separated populations.

## Figures and Tables

**Figure 1 animals-14-01167-f001:**
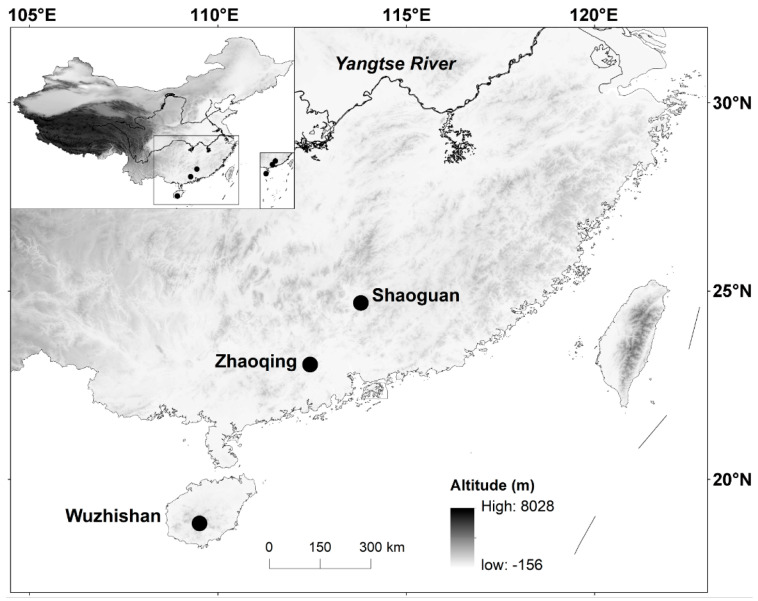
Map of China (top right corner), showing the three localities where we collected adult *T. sexlineatus*.

**Figure 2 animals-14-01167-f002:**
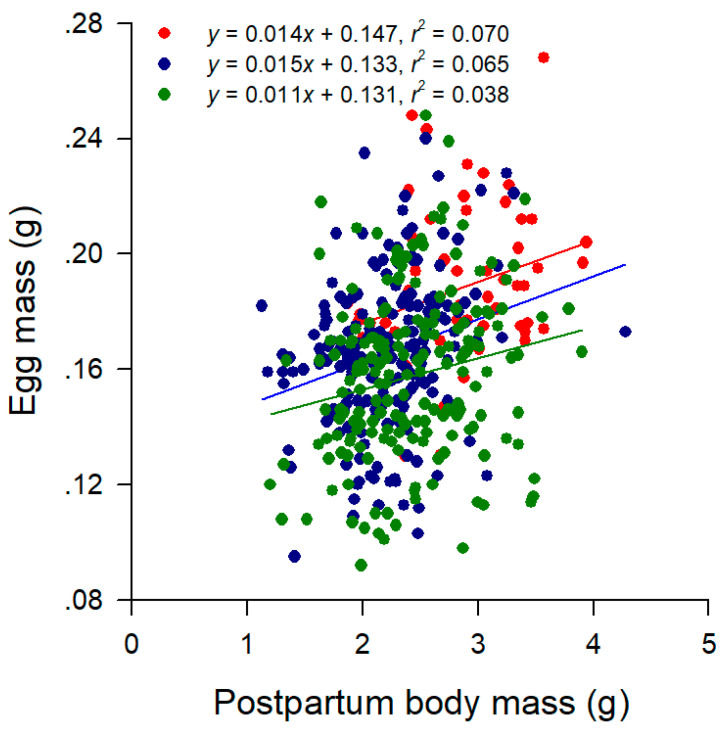
Linear regressions of egg mass on postpartum body mass in Wuzhishan (red dots and line), Shaoguan (blue dots and line), and Zhaoqing (green dots and line) females. Regression equations and coefficients are given the figure.

**Table 1 animals-14-01167-t001:** Descriptive statistics, expressed as mean ± SE and range, for post-laying female size and mass, egg mass, within-clutch variability [CV = 100 × (standard deviation/mean)] in egg mass, clutch size, clutch mass, and relative clutch mass in *T. sexlineatus* from the Wuzhishan, Shaoguan, and Zhaoqing populations. *N* = the number of females laying eggs.

Clutches	*N*	Postpartum Female Size and Mass		Egg Mass and Within-Clutch Variability		Clutch Size	Clutch Mass (g)	Relative Clutch Mass
		SVL (mm)	Body Mass (g)	Egg Mass (g)	CV of Egg Mass			
**Wuzhishan**								
First clutch	19	57.7 ± 0.8 49.6–64.2	2.7 ± 0.1 1.9–3.5	0.18 ± 0.003 0.16–0.21	4.6 ± 0.6 0.79–11.6	2.9 ± 0.09 2–4	0.53 ± 0.02 0.35–0.71	0.21 ± 0.01 0.13–0.29
Second clutch	19	56.6 ± 0.8 49.6–64.2	2.9 ± 0.1 2.0–3.9	0.20 ± 0.008 0.13–0.27	6.3 ± 2.2 0.70–39.4	2.8 ± 0.2 2–4	0.55 ± 0.04 0.34–0.88	0.19 ± 0.01 0.11–0.32
Third clutch	18	57.2 ± 0.7 49.6–64.2	2.9 ± 0.1 2.0–3.9	0.18 ± 0.007 0.13–0.24	4.2 ± 0.5 1.4–8.7	2.9 ± 0.1 2–4	0.53 ± 0.03 0.36–0.82	0.18 ± 0.05 0.15–0.22
Fourth clutch	4	58.7 ± 2.0 55.0–64.2	2.9 ± 0.07 2.7–3.0	0.18 ± 0.02 0.14–0.21	2.0 ± 0.4 0.90–2.8	2.5 ± 0.3 2–3	0.45 ± 0.05 0.37–0.59	0.15 ± 0.02 0.14–0.20
Fifth clutch	1	64.2	2.3	0.22	5.8	2	0.44	0.19
**Shaoguan**								
First clutch	104	51.2 ± 0.2 44.5–57.9	2.1 ± 0.04 1.1–4.3	0.16 ± 0.002 0.095–0.23	5.2 ± 0.5 0.35–21.9	2.5 ± 0.06 1–4	0.41 ± 0.01 0.17–0.66	0.20 ± 0.006 0.078–0.48
Second clutch	53	52.6 ± 0.4 44.6–57.8	2.3 ± 0.06 1.4–3.3	0.17 ± 0.004 0.10–0.24	3.5 ± 0.7 0.36–21.1	2.1 ± 0.08 1–3	0.34 ± 0.01 0.10–0.60	0.15 ± 0.007 0.04–0.31
Third clutch	29	53.2 ± 0.4 48.7–56.7	2.4 ± 0.07 1.9–3.2	0.17 ± 0.005 0.11–0.21	4.8 ± 0.9 0.40–14.8	2.1 ± 0.1 1–3	0.35 ± 0.02 0.12–0.59	0.14 ± 0.009 0.059–0.28
Fourth clutch	13	53.4 ± 0.4 51.6–55.6	2.4 ± 0.1 1.7–3.2	0.17 ± 0.009 0.11–0.23	4.3 ± 1.8 0.96–10.7	2.1 ± 0.1 1–3	0.35 ± 0.03 0.11–0.46	0.15 ± 0.01 0.059–0.21
Fifth clutch	4	54.6 ± 0.4 53.8–55.3	2.8 ± 0.1 2.6–3.0	0.19 ± 0.01 0.17–0.23	0.38	2.0 ± 0.4 1–3	0.38 ± 0.07 0.19–0.52	0.14 ± 0.02 0.069–0.17
**Zhaoqing**								
First clutch	78	50.8 ± 0.3 46.3–59.3	2.3 ± 0.05 1.2–3.6	0.16 ± 0.003 0.11–0.22	5.2 ± 0.5 0.34–17.6	2.6 ± 0.08 2–4	0.40 ± 0.01 0.21–0.74	0.18 ± 0.006 0.074–0.38
Second clutch	76	53.5 ± 0.3 48.4–61.4	2.6 ± 0.06 1.3–3.9	0.16 ± 0.003 0.09–0.24	4.6 ± 0.6 0.38–23.8	2.4 ± 0.07 1–4	0.37 ± 0.01 0.18–0.78	0.15 ± 0.005 0.067–0.27
Third clutch	33	54.4 ± 0.6 48.2–60.3	2.5 ± 0.07 1.6–3.4	0.16 ± 0.006 0.10–0.25	4.8 ± 0.9 0.47–16.4	2.3 ± 0.1 1–4	0.37 ± 0.02 0.19–0.68	0.15 ± 0.008 0.082–0.27
Fourth clutch	10	55.5 ± 0.8 51.5–59.2	2.5 ± 0.1 1.9–3.1	0.16 ± 0.008 0.12–0.20	6.6 ± 1.5 1.7–16.5	2.5 ± 0.2 2–4	0.40 ± 0.04 0.25–0.73	0.16 ± 0.02 0.11–0.33
Fifth clutch	0	—	—	—	—	—	—	—

**Table 2 animals-14-01167-t002:** Results of two-way ANOVA (female SVL, egg mass, and CV of egg mass) and ANCOVA (female postpartum body mass, clutch size, and clutch mass using female SVL as the covariate, and relative clutch mass using female postpartum body mass as the covariate) with population origin and clutch order as the fixed factors on the first three clutches. Populations and clutches with different superscripts differ significantly (Tukey’s post hoc test, α = 0.05, a > b > c). WZS, SG, and ZQ indicate the Wuzhishan, Shaoguan, and Zhaoqing populations, respectively. F, S, and T indicate the first, second, and third clutches, respectively.

	Population Origin	Clutch Order	Population × Clutch Interaction
Snout vent length	*F*_2, 420_ = 49.07, *p* < 0.0001 WZS ^a^, SG ^b^, ZQ ^b^	*F*_2, 420_ = 19.70, *p* < 0.0001 F ^b^, S2 ^a^, T3 ^a^	*F*_2, 420_ = 1.92, *p* = 0.105
Postpartum body mass	*F*_2, 419_ = 5.87, *p* < 0.01 WZS ^a^, SG ^b^, ZQ ^b^	*F*_2, 419_ = 2.81, *p* = 0.062	*F*_2, 419_ = 0.94, *p* = 0.441
Egg size	*F*_2, 420_ = 23.81, *p* < 0.0001 WZS ^a^, SG ^b^, ZQ ^c^	*F*_2, 420_ = 1.66, *p* = 0.191	*F*_2, 420_ = 1.25, *p* = 0.290
CV of egg mass	*F*_2, 343_ = 0.33, *p* = 0.721	*F*_2, 343_ = 0.149, *p* = 0.861	*F*_2, 343_ = 1.04, *p* = 0.385
Clutch size	*F*_2, 419_ = 7.42, *p* < 0.001 WZS ^a^, SG ^c^, ZQ ^b^	*F*_2, 419_ = 14.14, *p* < 0.0001 F ^a^, S ^b^, T ^b^	*F*_2, 372_ = 0.91, *p* = 0.446
Clutch mass	*F*_2, 419_ = 17.45, *p* < 0.0001 WZS ^a^, SG ^b^, ZQ ^b^	*F*_2, 372_ = 11.60, *p* < 0.0001 F ^a^, S ^b^, T ^b^	*F*_2, 372_ = 1.47, *p* = 0.209
Relative clutch mass	*F*_2, 419_ = 28.80, *p* < 0.0001 WZS ^a^, SG ^b^, ZQ ^b^	*F*_2, 419_ = 6.58, *p* < 0.01 F ^a^, S ^b^, T ^b^	*F*_2, 372_ = 1.11, *p* = 0.349

## Data Availability

Data are contained within the article.

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
