# Peer review of "Species-Specific Seasonal Shifts in Reproductive Allocation in the Southern Grass Lizard, Takydromus sexlineatus (Lacertidae)"

_animals, 2024, doi:10.3390/ani14081167_

Round 1

Reviewer 1 Report

Comments and Suggestions for Authors

The authors present a fine study, so my only important concern is that, for statistics, the authors should use mixed models with female ID as a random factor. Given that for each female the authors have several data (clutches), there is a pseudoreplication problem easily solved with a mixed model.

- Title. I propose a shorter title, such as: "Species-specific Pattern of Seasonal Shifts in Reproductive Allocation in the Southern Grass Lizard,  Takydromus sexlineatus (Lacertidae"

- Simple summary: In line 23 the authors say that "Our data validate the hypothesis tested", but they do not present either hypothesis. Please, change the first sentence to "We used a common garden to collect data on female reproductive traits from three populations of the southern grass lizard Takydromus sexlineatus, testing the hypothesis that geographically separated populations should share a species-specific pattern of seasonal shifts in reproductive allocation."

- Abstract: Please, delete the reference to Smith and Fretwell’s (1974). It is not adequate to cite references in the abstract.

- The Introduction might be improved with stronger conceptual sentences and clearer definitions of the hypotheses. Still, I find the Introduction is fine.

- P.3 L. 124: an erroneous symbol just after “28”. The same symbol is in L. 198.

- Regarding the increase of SVL along the study, please, notice that females have ad libitum conditions of food and thermal requirements, which are not fulfilled in the field. Please, add a Discussion on how lab conditions might affect some of the results.

Author Response

Reviewer 1

The authors present a fine study, so my only important concern is that, for statistics, the authors should use mixed models with female ID as a random factor. Given that for each female the authors have several data (clutches), there is a pseudoreplication problem easily solved with a mixed model.

>> Thanks for your suggestion. It is better to use mixed models with female ID as a random factor to analyse data, so that the family (maternal) effect can be examined. Unfortunately, our data do not allow to do so because not all clutches laid by individual females throughout the breeding season were normal (without any infertile eggs and/or abnormal eggs with condensed yolk)

- Title. I propose a shorter title, such as: “Species-specific Pattern of Seasonal Shifts in Reproductive Allocation in the Southern Grass Lizard, Takydromus sexlineatus (Lacertidae”

>> Thanks for your suggestion. The title has been shortened by 10 words and now changed to “Species-specific Pattern of Seasonal Shifts in Reproductive Allocation in the Southern Grass Lizard, Takydromus sexlineatus (Lacertidae)”

- Simple summary: In line 23 the authors say that “Our data validate the hypothesis tested”, but they do not present either hypothesis. Please, change the first sentence to “We used a common garden to collect data on female reproductive traits from three populations of the southern grass lizard Takydromus sexlineatus, testing the hypothesis that geographically separated populations should share a species-specific pattern of seasonal shifts in reproductive allocation.”

>> Done. Thanks for your suggestion

- Abstract: Please, delete the reference to Smith and Fretwell’s (1974). It is not adequate to cite references in the abstract.

>> Done. Thanks for your suggestion

- The Introduction might be improved with stronger conceptual sentences and clearer definitions of the hypotheses. Still, I find the Introduction is fine.

>> Thanks for your positive comments

- P.3 L. 124: an erroneous symbol just after “28”. The same symbol is in L. 198.

>> Corrected. Thanks

- Regarding the increase of SVL along the study, please, notice that females have ad libitum conditions of food and thermal requirements, which are not fulfilled in the field. Please, add a Discussion on how lab conditions might affect some of the results.

>> Thanks for your suggestion. We added the following sentence at the end of the first paragraph of the Discussion

“It is worth noting that the increase of SVL and optimal reproductive allocation along this study were achieved in females having ad libitum conditions of food and thermal requirements, which are very unlikely to be fulfilled in the field due to various limitations in resource acquisition.”

Reviewer 2 Report

Comments and Suggestions for Authors

Thank you for sending me an interesting manuscript for review. From the works cited, it is clear that your team has been working on similar topics for a long time, so you have experience with the methodology and analysis of the collected data. I do not in any way question your chosen methodology or interpretation of the results, however, I have a few comments on the structure of your paper and a few additional questions.
The title of the article is already a bit difficult to understand. It does contain all the essential information about the content of the article, but perhaps to an unnecessary extent. Please consider modifying the title to be shorter and convey the most important results (or
objectives) of your research.
I have a similar problem with the results and data interpretation. Everything is precisely described here, but the reader soon gets lost in the repetitive concepts. At the end of the introduction, I missed setting out clear objectives and hypotheses to which the results and subsequent discussion would relate. I do not claim that the aims of the article are not stated here or that the results do not correspond to them. It's just that it's hard to notice them and realize what is being discussed in the following paragraphs. Also, it would have been useful to clearly state which seven traits will be measured in addition to the objectives. Again, this can be ascertained from the following text, but after a more complicated search.
For the data analysis, I wonder why you counted three clutches when you have four clutches from all sites. Is this an established methodology or did you have another reason?
A map with the locations where the lizards were collected might be useful to add to the populations studied, especially for readers who don't live in China. You do have coordinates in your methodology, but the map is more illustrative.

Figure 1 is rather vaguely described. Although in the caption you state the model it is based on and which populations it applies to, it is not clear why you show this particular result. Again, after searching the text and discussion, the reader will find that this is one of the important results. The caption should be self-explanatory if possible. Alternatively, consider adding another Figure with an interesting result, for example, including a result that did not come out conclusive, contrary to expectations.
On the other hand, I appreciate the clear table with the collected data, the very well described methodology and the summary of basic information about the studied species.
Your research is very interesting and useful, so I hope that my comments will help to improve your paper.

Author Response

Reviewer 2

Thank you for sending me an interesting manuscript for review. From the works cited, it is clear that your team has been working on similar topics for a long time, so you have experience with the methodology and analysis of the collected data. I do not in any way question your chosen methodology or interpretation of the results, however, I have a few comments on the structure of your paper and a few additional questions.

>> Thanks for your positive comments

The title of the article is already a bit difficult to understand. It does contain all the essential information about the content of the article, but perhaps to an unnecessary extent. Please consider modifying the title to be shorter and convey the most important results (or objectives) of your research.

>> Thanks for your suggestion. The title has been shortened by 10 words and now changed to “Species-specific Pattern of Seasonal Shifts in Reproductive Allocation in the Southern Grass Lizard, Takydromus sexlineatus (Lacertidae)”

I have a similar problem with the results and data interpretation. Everything is precisely described here, but the reader soon gets lost in the repetitive concepts. At the end of the introduction, I missed setting out clear objectives and hypotheses to which the results and subsequent discussion would relate. I do not claim that the aims of the article are not stated here or that the results do not correspond to them. It’s just that it’s hard to notice them and realize what is being discussed in the following paragraphs. Also, it would have been useful to clearly state which seven traits will be measured in addition to the objectives. Again, this can be ascertained from the following text, but after a more complicated search.

>> We rewrote the last sentence related to the hypothesis tested. The end of the Introduction has been changed to “We hypothesize that a species-specific pattern should be shared by geographically separated populations, whereas a population-specific should differ among geographically separated populations”

>> The seven measured female traits were noted in the Introduction. Hope it is easier to follow

For the data analysis, I wonder why you counted three clutches when you have four clutches from all sites. Is this an established methodology or did you have another reason?

>> We had five clutches from the Hainan and Shaoguan populations, and four clutches from the Zhaoqing population. We only included the first three clutches to test the effects of population, clutch order and their interaction because only four Wuzhishan (Hainan) females laid their fourth clutch, but the sample size was too small to perform reliable statistical analyses

A map with the locations where the lizards were collected might be useful to add to the populations studied, especially for readers who don’t live in China. You do have coordinates in your methodology, but the map is more illustrative.

>> Thanks for your suggestion. We added a figure to show the three localities where we collected adult T. sexlineatus. Please see Figure 1 in the revised manuscript for details

Figure 1 is rather vaguely described. Although in the caption you state the model it is based on and which populations it applies to, it is not clear why you show this particular result. Again, after searching the text and discussion, the reader will find that this is one of the important results. The caption should be self-explanatory if possible. Alternatively, consider adding another Figure with an interesting result, for example, including a result that did not come out conclusive, contrary to expectations.

>> This figure (now Figure 2 in the revised manuscript) shows that female postpartum mass explains a small but significant proportion of variation in egg mass in T. sexlineatus

On the other hand, I appreciate the clear table with the collected data, the very well described methodology and the summary of basic information about the studied species.

>> Thanks for your suggestion. I don’t quite understand the necessity of the table you suggested. Hope our tables (Table 1 and Table 2) are self-explanatory and easy to follow

Your research is very interesting and useful, so I hope that my comments will help to improve your paper.

>> Thank you again for your positive comments

Reviewer 3 Report

Comments and Suggestions for Authors

The research developed by the authors is of great interest because the study of variations in the reproductive characteristics of various species of lizards has been neglected. However, the manuscript can be improved in several aspects. In attachment are comments, changes and suggestions for the authors 

Comments on the Quality of English Language

Minor editing of English language required

Author Response

Reviewer 3

The research developed by the authors is of great interest because the study of variations in the reproductive characteristics of various species of lizards has been neglected. However, the manuscript can be improved in several aspects. In attachment are comments, changes and suggestions for the authors.

>> Thanks for your positive comments and suggestions

Line 55: It would be important to point out the latitudinal and altitudinal variations with their respective references

>> Thanks for your suggestion, but it could not be necessary to point out latitudinal and altitudinal variations in the species where geographical variation in reproductive traits has been examined for two reasons. First, detailed spatial coordinate information is not available for all species so far studied. Second, even if available for some species, we need a large table to present these coordinate data but such an effort is definitely beyond the scope of this paper

Lines 56-57: Consult more papers on reproductive characteristics, mainly in Xenosauridae and Gekkonidae that allow this assertion

>> Rarely can we find papers on geographical variation in reproductive characteristics in these two lizard families

Line 121: Was this ratio for mating purposes? If so, explain

>> We changed the sentence to “Following a previous study on P. muralis for mating purpose [35], we housed 9-12 lizards (with a female-to-male ratio of approximately 2 to 1) from the same population in each ……”

Line 128: Were the females collected in the field already pregnant? Was copulation and the females fertilized in the laboratory? This part of the method is confusing, please detail

>> We changed the sentence to “We collected adult males and females with various sized yolking follicles in late March 2015 and 2016 from ……”

Line 132: How did the authors determine the presence of eggs in the oviduct?

>> We added a sentence (We palpated females at 3-d intervals for the presence of eggs in the oviduct) to show how we determine the presence of eggs in the oviduct. Thanks

Table 1: Delete, It is just anecdotal data, without statistical significance

>> Thanks for your suggestion, but it is better to show the whole picture of reproductive events in an egg-laying (breeding) season. We excluded the fourth and fifth clutches from analyses because the sample size in at least one population was too small to perform reliable statistical analyses.

References: I am surprised not to see any work by Laurie Vitt in the references, since he was one of the initiators of this type of studies (see https://samnoblemuseum.ou.edu/staff/laurie-j-vitt/)

>> Previous studies including those by Laurie Vitt and his team rarely use data covering the whole breeding season and from multiple populations to compare reproductive traits among populations. We only found one paper (Garda et al., 2012) comparing reproductive traits between polychrotid lizards living in two contrasting environments in the above link. We cited the following paper by Vitt’s team in the revised manuscript

Garda, A.A.; Tavares-Bastos, L.; Costa, G.C.; França, F.G.R.;  Giugliano, L.G.;  Leite, G.S.; Mesquita, D.O.;  Nogueira, C.; Vasconcellos, M.M.; Vieira, G.H.C.; Vitt, L.J.;  Werneck, F.P.; Wiederhecker, H.C.; Colli, G.R. Reproduction, body size, and diet of Polychrus acutirostris (Squamata, Polychrotidae) in two contrasting environments in Brazil. J. Herpetol. 2012, 46, 2-8.

Round 2

Reviewer 1 Report

Comments and Suggestions for Authors

None.

Author Response

Ms. Crystal Zhang

Section Managing Editor

Animals

Wenzhou, April 4, 2024

Species-specific Seasonal Shifts in Reproductive Allocation in the Southern Grass Lizard, Takydromus sexlineatus (Lacertidae)

ID: animals-2908368

Dear Ms. Zhang,

Thank you for sending a new decision letter on the above manuscript on April 4, 2024. It is a pleasure to hear that our manuscript now is still in a status of Minor Revision. As the senior corresponding author, I revised once again the manuscript, paying particular attention to the attached PDF file with annotations.

Below you will find the new decision letter including all editor comments. We have indicated our responses and changes in a blue colour following each comment.

We are waiting for further comments from you, and would greatly appreciate it if the manuscript can be improved at the editorial stage. Thank you again for considering our manuscript.

Sincerely yours,

Xiang Ji

Professor in Ecology & Zoology

Dear Professor Ji,

Thank you again for your manuscript submission:

Manuscript ID: animals-2908368

Title: Species-specific Pattern of Seasonal Shifts in Reproductive Allocation in the Southern Grass Lizard, Takydromus sexlineatus (Lacertidae)

Authors: Cai-Feng Wang, Yu Du, Kun Guo, Xiang Ji *

Received: 23 Feb 2024

E-mails: wangcaifeng@stu.wzu.edu.cn, yudu@hntou.edu.cn,

guokun8808@wzu.edu.cn, xji@wzu.edu.cn

Herpetology

https://www.mdpi.com/journal/animals/sections/Herpetology

Your manuscript has been reviewed by experts in the field. Please find your manuscript with the referee reports at this link:

https://susy.mdpi.com/user/manuscripts/resubmit/dbc29faca6613a7287b977c508549ecd

(I) Please revise your manuscript according to the referees’ comments and upload the revised file within 5 days (by 9 April 2024).

>> Done. Thanks

(II) Please use the version of your manuscript found at the above link for your revisions.

>> Done. Thanks

(III) Please check that all references are relevant to the contents of the manuscript.

>> Done. Thanks

(IV) Any revisions to the manuscript should be highlighted, such that any changes can be easily reviewed by editors and reviewers.

>> Done. Thanks

(V) Please provide a short cover letter detailing your changes for the editors’ and referees’ approval.

>> Done. Thanks

(VI) If the reviewer(s) recommended references, please critically analyze them to ensure that their inclusion would enhance your manuscript. If you believe these references are unnecessary, you should not include them.

>> Checked. Thanks

If one of the referees has suggested that your manuscript should undergo extensive English revisions, please address this issue during revision. We propose that you use one of the editing services listed at

https://www.mdpi.com/authors/english or have your manuscript checked by a

colleague fluent in English writing.

>> No need to undergo extensive English revisions

Please do not hesitate to contact us if you have any questions regarding the revision of your manuscript or if you need more time. We look forward to hearing from you soon.

Kind regards,

Ms. Crystal Zhang

Section Managing Editor

MDPI Branch Office, Wuhan

Animals Editorial Office

Tel.: +027-59972063

E-mail: animals@mdpi.com

https://www.mdpi.com/journal/animals/
